# Association of menopausal status with body composition and anthropometric indices in Korean women

**Bum Ju Lee** [1]*, **Jaeuk U. Kim**[1], **Sanghun Lee**[2]

1 Digital Health Research Division, Korea Institute of Oriental Medicine, Daejeon, Republic of Korea, 2 KM Data Division, Korea Institute of Oriental Medicine, Daejeon, Republic of Korea

* bjlee@kiom.re.kr

## Abstract

### Background

Menopause induces various health problems and is associated with obesity, but the association between menopausal status and obesity is unclear due to several confounding factors, such as aging and reduced physical activity. The objective of this study was to examine the association of menopausal status with anthropometric indices and body composition indices in South Korean women.

### Methods

In this cross-sectional study, a total of 734 subjects (297 postmenopausal women, 437 premenopausal women) from five university hospitals in South Korea were included. A binary logistic regression analysis was performed to examine the association of menopause with anthropometric indices and body composition indices.

### Results

Height, body mass index, waist-to-height ratio, waist-to-hip ratio, and neck, armpit, chest, rib, waist, iliac, and hip circumferences were associated with menopausal status in the crude analysis, but these associations disappeared in the adjusted models. Among the body composition indices, menopausal status was strongly associated with total body water, skeletal muscle mass, body fat mass, and body fat percentage in the crude analysis. However, the associations with body fat mass and body fat percentage disappeared in the adjusted models. Only the associations with total body water and skeletal muscle mass remained in the adjusted models.

### Conclusion

Most of the anthropometric indices and body composition indices were not associated with menopausal status, but total body water and skeletal muscle mass were significantly lower in postmenopausal women than in premenopausal women.

**Data Availability Statement:** The datasets generated and analyzed during this study are not publicly available due to the confidentiality policy of the Korea Institute of Oriental Medicine (KIOM). The data was collected from five hospitals

(Dongshin University Korean Medicine Hospital, Pusan National University Korean Hospital, Gachon University Gill Hospital, Dongguk University Medical Center, and Daejeon University Oriental Medicine Hospital) and is still being collected. The data underlying the results presented in the study are available from KIOM (https://kiom.re.kr/) by emailing to thkim@kiom.re.kr (Dr. Taehong Kim, data manager in KIOM).

**Funding:** Initials of the authors who received each award: JUK and SHL This study was supported by the Korea Institute of Oriental Medicine (KIOM; Grant no.: KSN2022130 and KSN2021110) funded by the Korean government. The funders had no role in study design, data collection and analysis, decision to publish, or preparation of the manuscript.

**Competing interests:** The authors have declared that no competing interests exist.

## Introduction

Generally, menopause is known as an endocrinological transition from reproductive to postreproductive life and is characterized by the permanent cessation of menses resulting from a reduction in ovarian follicles in women [1–3]. Menopause induces health problems such as cardiovascular diseases, lipid profile disturbances, metabolic disturbances, psychological stress, and central adiposity, leading to increased mortality in middle-aged and elderly women [1, 2, 4]. The known risk factors for early or fast natural menopause are current smoking status, lower educational attainment, separated/widowed/divorced marital status, unemployment, history of heart disease, obesity, psychosocial stress, weight reduction, irregular menstrual cycles, and ethnicity [5–7].

Over an extended period, many studies have examined the association between menopausal status and obesity or adiposity. However, the associations between menopausal status and obesity and weight gain remain controversial [3]. Several studies have concluded that menopause and menopausal transition are associated with increases in obesity and anthropometric indices [6, 8–11] due to hypoestrogenic effects because estrogens affect gluteofemoral adiposity [12] or due to the effect of bioavailable testosterone [13]. However, some studies have shown that obesity and anthropometric indices are not associated with menopausal status [3, 12, 14–17], suggesting that the changes in anthropometric and body composition indices are due to aging or reductions in physical activity, not menopausal status [3, 14, 16]. For example, natural menopause was not associated with elevated body mass index (BMI) in ethnically diverse women who were middle-aged, and a powerful indicator of adiposity in middle-aged women was low physical activity [14]. Chronological aging has been implicated as the main cause of obesity because adiposity similarly increases in both men and women of the same age [3, 16]. Another study concluded that the observed decreases in total and trunk lean mass were more strongly associated with menopause, while body fat mass and overall adiposity were more strongly related to aging [12]. These conflicting results on the association between menopausal status and obesity may result in confusion. Therefore, the objective of the present cross-sectional study was to examine the association of menopausal status with anthropometric indices and body composition indices in South Korean women and to determine whether menopausal status affects obesity or adiposity. Our findings may provide fundamental information on the associations between menopause and obesity or adiposity in Korean women.

## Methods

### Subjects and data sources

The data used in the present study were collected from five university hospitals, Dongshin University Korean Medicine Hospital (Naju city), Pusan National University Korean Hospital (Yangsan city), Gachon University Gill Hospital (Incheon city), Dongguk University Medical Center (Goyang city), and Daejeon University Oriental Medicine Hospital (Daejeon city), from 2020 to 2022 in the Republic of Korea. All participants in this study signed informed consent forms, and the protocols were approved by the Institutional Review Board of the Dongshin University Korean Medicine Hospital (NJ-IRB-003), Pusan National University Korean Hospital (PNUKHIRB-2020005), Gachon University Gill Hospital (GIRB-20-113), Dongguk University Medical Center (DUIOH-2020-04-002), and Daejeon University Oriental Medicine Hospital (JDSKH-20-BM-07). This study was performed in accordance with the Declaration of Helsinki, and all methods were conducted in accordance with the relevant guidelines and regulations.

A total of 2,178 subjects participated in this study from 2020 to 2022. Among them, we selected 1,444 female subjects aged 35 to 65 years and then excluded 710 subjects with missing data on anthropometric indices such as height and weight; body composition indices such as body water and fat mass; and sociodemographic characteristics such as smoking status, alcohol use, blood pressure levels, and physical activity levels. Finally, 734 subjects (297 postmenopausal women, 437 premenopausal women) were included in our experiments. Fig 1 presents

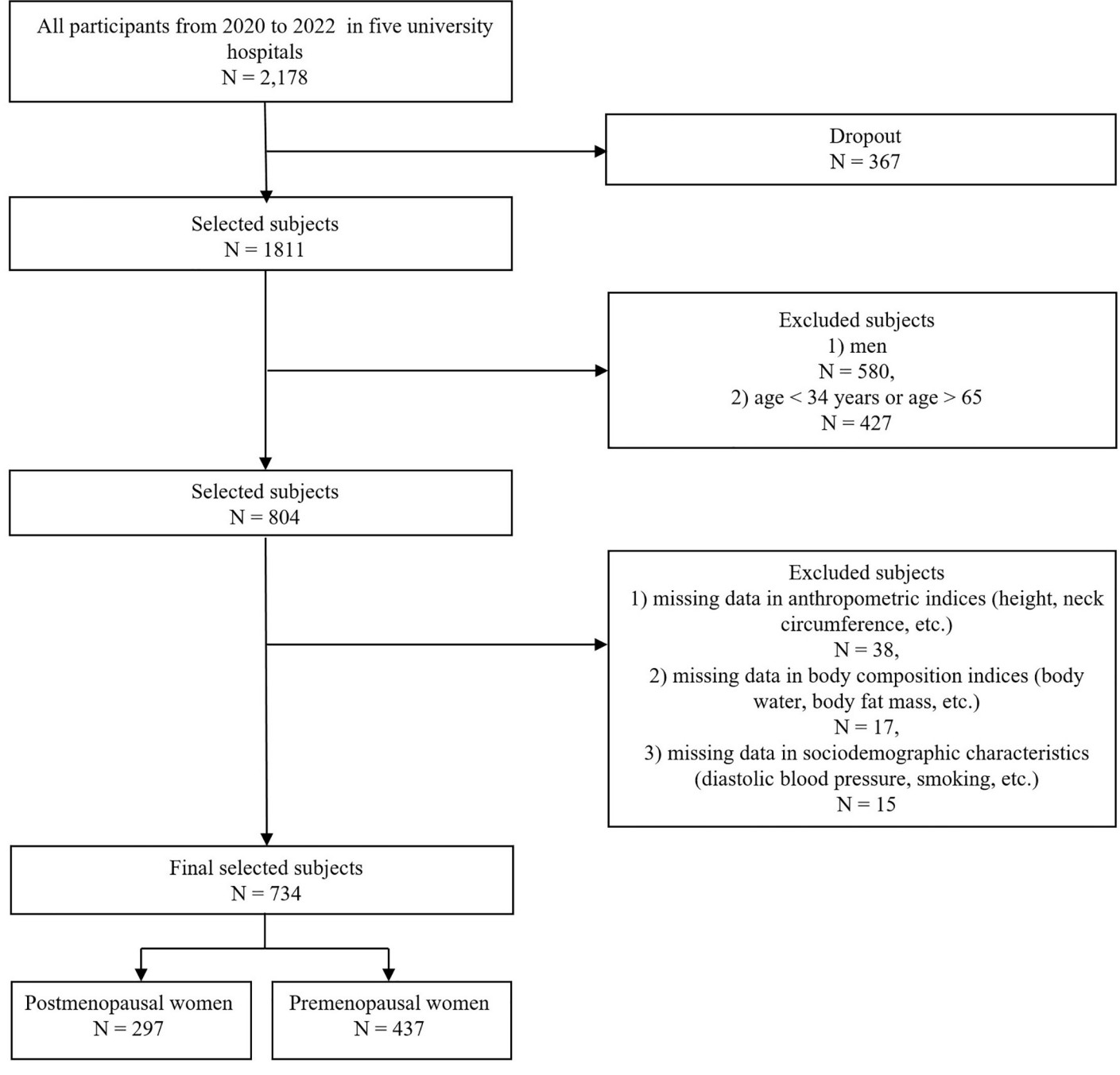

**Fig 1. Sample selection procedure in this study.**

the sample selection procedure, including the detailed inclusion and exclusion criteria. All the data used in our study were anonymized.

## Definition of postmenopausal and premenopausal women

Information on the postmenopausal and premenopausal women was obtained by well-trained technicians or observers through face-to-face health interviews. Women who answered "yes" to the question "Have you reached menopause?" were placed in the postmenopausal group. Women who answered "no" to this question were placed in the premenopausal group.

## Anthropometry and body composition measurements

Demographic characteristics such as age, occupation, education levels, marital status, alcohol use, and smoking status were documented by administering a questionnaire to all subjects. The systolic and diastolic blood pressures of the subjects were measured by a sphygmomanometer based on the oscillometric method (ACCUNIQ BP 850, Selvas Healthcare, Inc., Daejeon, Korea). Height and weight were measured to the nearest 0.1 cm and 0.1 kg, respectively, while the participants wore light clothes according to standard procedures performed by well-trained medical staff (Inbody BSM 370, Biospace Co. Ltd., Seoul, Korea). BMI was calculated as weight divided by height squared (kg/m$^2$). Body composition indices, including total body water, skeletal muscle mass, body fat mass, and body fat percentage, were measured by multifrequency bioelectrical impedance analysis (BIA) devices (Inbody BWA 2.0, Biospace Co. Ltd., Seoul, Korea) while the subjects laid on a bed with clamp-type sensors on both their wrists and ankles according to the measurement protocols and the procedures provided by the manufacturers. After resting for 5 minutes, the device began to measure and automatically calculate body composition. For specific anthropometric indices, well-trained physicians or observers used measuring tape to measure the circumferences of eight specific body sites, including the forehead, neck, armpit, chest, rib, waist, iliac, and hip, to the nearest 0.1 cm; all of the participants wore lightweight clothing based on standardized protocols. For example, forehead and neck circumferences were measured at the glabellar and occiput levels and at the thyroid cartilage and cricoid cartilage levels. Detailed descriptions of the eight body sites and methods have been provided in previous studies [18–21]. Additionally, the waist-to-height (WHtR) and waist-to-hip (WHR) ratios were calculated.

## Statistical analysis

All the statistical analyses were performed with SPSS Statistics, version 23.0 (IBM Corp., Armonk, NY, USA). A significance level of 0.05 was applied in all analyses. Categorical variables were examined by the chi-square test, and continuous variables were examined by the independent two-sample t test. In the crude analysis and adjustment analysis of potential confounders, a binary logistic regression analysis was performed to examine the associations of menopausal status with the anthropometric indices and body composition indices after standardization of the data. Model 1 included one variable and covariate (age) because anthropometry and body composition may change according to chronological age. Model 2 included one variable and several covariates (age, location, occupation, education level, marital status, smoking status, alcohol use, high-level activity participation, and SBP and DBP levels) for adjustment. Odds ratios are shown with 95% confidence intervals (CIs) and p values for all models.

## Results

### Basic characteristics of the postmenopausal and premenopausal groups

Table 1 shows the overall demographic characteristics between the postmenopausal and premenopausal groups. A total of 734 women were included in the analysis (297 postmenopausal and 437 premenopausal women). The mean (standard deviation) age of the subjects was 56.7 (5.36) years in the postmenopausal group and 44.3 (5.29) years in the premenopausal group. Diastolic (DBP) and systolic (SBP) blood pressure levels and weekly high-level activity participation were greater in the postmenopausal group than in the premenopausal group. Location (residential area), occupation, education level, marital status, and alcohol use strongly differed between the postmenopausal and premenopausal groups.

### Association of menopause with anthropometry and body composition

Table 2 shows the associations of menopausal status with anthropometric indices and body composition indices. According to the crude analysis, all the anthropometric indices except for weight, forehead circumference, and hip circumference were associated with menopause, and height, BMI, WHtR, WHR, and neck, rib, and iliac circumferences were strongly associated with menopause. However, these associations disappeared in Model 1 after we adjusted for age and in Model 2 after we adjusted for the following covariates: age, location, occupation, education level, marital status, smoking status, alcohol use, high-level activity participation, and SBP and DBP levels.

Among the body composition indices, menopause was strongly associated with total body water (OR = 1.56 [95% CI 1.33–1.84], $p < 0.001$), skeletal muscle mass (OR = 1.61 [1.36–1.90], $p < 0.001$), body fat mass (OR = 0.81 [0.70–0.94], $p = 0.006$), and body fat percentage (OR = 0.64 [0.54–0.75], $p < 0.001$) in the crude analysis. However, the associations between body fat mass and body fat percentage disappeared in Models 1 and 2. In Model 1, total body water (adj. OR = 1.40 [1.09–1.81], adj. $p = 0.010$) and skeletal muscle mass (adj. OR = 1.41 [1.09–1.82], adj. $p = 0.009$) maintained statistical significance. Additionally, in Model 2, total body water (adj. OR = 1.51 [1.13–2.03], adj. $p = 0.006$) and skeletal muscle mass 1.51 [1.13–2.02], adj. $p = 0.006$) were strongly associated with menopausal status. The magnitudes of the associations of menopause with total body water and skeletal muscle mass were similar in the crude analysis, Model 1, and Model 2.

## Discussion

In this study, we examined the associations of anthropometric indices and body composition indices with menopausal status because the associations have remained unclear. We found that although all the anthropometric indices, except for weight and forehead and hip circumferences, were significantly associated with menopause, all associations disappeared in the adjusted models. Among the body composition indices, total body water and skeletal muscle mass were strongly associated with menopause in all the crude and adjusted models, but body fat mass and fat mass percentage were not related to menopause in the adjusted models.

However, the associations between menopausal status and obesity and anthropometric indices are still debatable. Donato et al. [8] examined the associations of menopausal status with waist circumference (WC) and the WHR in Brazil by a cross-sectional study of a representative population-based cohort. They argued that premenopausal women had greater height and smaller WC and WHR than postmenopausal women did. Additionally, postmenopausal women had a high risk of central adiposity, independent of several confounders,

**Table 1. General characteristics of the postmenopausal and premenopausal groups.**

| Category | Variable | Postmenopausal | Premenopausal | p value |
|---|---|---|---|---|
| | | Mean (SD) or frequency (%) | Mean (SD) or frequency (%) | |
| | Subjects (n) | 297 | 437 | |
| | AGE (years) | 56.7 (5.36) | 44.3 (5.29) | <0.001 |
| | Diastolic BP (mmHg) | 73.3 (9.21) | 71.4 (9.21) | 0.006 |
| | Systolic BP (mmHg) | 120 (13.4) | 115 (12.4) | <0.001 |
| | High level activity per week (frequency) | 0.99 (1.62) | 0.73 (1.37) | 0.020 |
| | Number of pregnancies (frequency) | 2.79 (1.46) | 2.23 (1.33) | <0.001 |
| | Number of childbirths (frequency) | 1.88 (0.95) | 1.66 (0.97) | 0.002 |
| | Height (cm) | 158 (5.30) | 161 (5.01) | <0.001 |
| Anthropometry | Weight (kg) | 58.1 (7.97) | 58.5 (8.41) | 0.577 |
| | Body mass index (BMI) | 23.3 (3.02) | 22.6 (3.15) | 0.003 |
| | Forehead circumference (cm) | 55.5 (1.96) | 55.7 (1.88) | 0.262 |
| | Neck circumference (cm) | 33.3 (2.28) | 32.8 (2.12) | 0.008 |
| | Armpit circumference (cm) | 88.7 (5.91) | 87.8 (6.29) | 0.046 |
| | Chest circumference (cm) | 91.4 (7.21) | 90.1 (6.98) | 0.011 |
| | Rib circumference (cm) | 80.3 (7.43) | 78.5 (7.49) | 0.001 |
| | Waist circumference (cm) | 83.4 (7.75) | 81.8 (7.99) | 0.010 |
| | Iliac circumference (cm) | 91.8 (6.62) | 90.3 (6.95) | 0.003 |
| | Hip circumference (cm) | 95.1 (5.12) | 95.8 (5.57) | 0.066 |
| | Waist-to-height ratio (WHtR) | 0.53 (0.05) | 0.51 (0.05) | <0.001 |
| | Waist-to-hip ratio (WHR) | 0.88 (0.06) | 0.85 (0.06) | <0.001 |
| Body composition | Total body water (l) | 27.5 (2.94) | 28.8 (3.12) | <0.001 |
| | Skeletal muscle mass (kg) | 19.8 (2.36) | 20.9 (2.50) | <0.001 |
| | Body fat mass (kg) | 21.0 (5.37) | 19.8 (5.65) | 0.006 |
| | Body fat percentage (%) | 35.6 (5.71) | 33.2 (5.55) | <0.001 |
| Location | Goyang city | 76 (10%) | 75 (10%) | <0.001 |
| | Daejeon city | 34 (4.6%) | 38 (5.2%) | |
| | Naju city | 84 (11%) | 112 (15%) | |
| | Incheon city | 49 (6.7%) | 68 (9.3%) | |
| | Yangsan city | 54 (7.4%) | 144 (20%) | |
| Occupation | Manager | 2 (0.3%) | 5 (0.7%) | <0.001 |
| | Professional worker | 50 (6.8%) | 110 (15%) | |
| | Office worker | 27 (3.7%) | 81 (11%) | |
| | Service worker | 31 (4.2%) | 29 (4%) | |
| | Sales worker | 9 (1.2%) | 8 (1.1%) | |
| | Farmer or fisher | 4 (0.5%) | 1 (0.1%) | |
| | Technical worker | 5 (0.7%) | 3 (0.4%) | |
| | Simple labor | 6 (0.8%) | 7 (1%) | |
| | Housewife | 159 (22%) | 186 (25%) | |
| | Other | 4 (0.5%) | 7 (1%) | |
| Education | Uneducated | 1 (0.1%) | 0 (0%) | <0.001 |
| | Elementary school | 6 (0.8%) | 0 (0%) | |
| | Middle school | 31 (4.2%) | 2 (0.3%) | |
| | High school | 117 (16%) | 69 (9.4%) | |
| | College | 35 (4.8%) | 83 (11%) | |
| | University | 87 (12%) | 239 (33%) | |
| | Graduate school | 20 (2.7%) | 44 (6%) | |

*(Continued)*

**Table 1.** (Continued)

| Category | Variable | Postmenopausal | Premenopausal | p value |
|---|---|---|---|---|
| | | Mean (SD) or frequency (%) | Mean (SD) or frequency (%) | |
| Marital status | Unmarried | 8 (1.1%) | 43 (5.9%) | <0.001 |
| | Married | 270 (37%) | 385 (53%) | |
| | Bereaved | 4 (0.5%) | 3 (0.4%) | |
| | Divorced | 15 (2%) | 6 (0.8%) | |
| Smoking | Yes | 10 (1.4%) | 21 (2.9%) | 0.342 |
| | No | 287 (39%) | 416 (57%) | |
| Alcohol use | Yes | 128 (17%) | 295 (40%) | <0.001 |
| | No | 169 (23%) | 142 (19%) | |

P values were derived from independent two-sample t tests. Numerical values are presented as the mean (SD: standard deviation), and categorical values are presented as frequencies (percentages).

such as age, hormone therapy, socioeconomic status, behavioral features, and BMI. Toth et al. [9, 10] tested the effect of menopausal status on central fat distribution and body composition in premenopausal and postmenopausal women using computed tomography (CT) and dual-energy X-ray absorptiometry (DEXA). They stated that both the menopausal transition and postmenopausal status were related to an increase in central adiposity, independent of age, BMI and total body fat mass. Sternfeld et al. [16] assessed the associations among age, menopausal status, obesity, and physical activity in a longitudinal study using a

**Table 2. Associations of menopausal status with anthropometric indices and body composition indices in Korean women.**

| Category | Variable | Crude | p | Model 1 | Adj. p | Model 2 | Adj. p |
|---|---|---|---|---|---|---|---|
| | | OR (95% CI) | | Adj. OR (95% CI) | | Adj. OR (95% CI) | |
| Anthropometry | Height | 1.81 (1.54–2.14) | <0.001 | 1.21 (0.93–1.56) | 0.149 | 1.26 (0.95–1.68) | 0.115 |
| | Weight | 1.04 (0.90–1.21) | 0.576 | 1.16 (0.90–1.50) | 0.245 | 1.18 (0.89–1.56) | 0.247 |
| | BMI | 0.80 (0.69–0.93) | 0.003 | 1.05 (0.82–1.35) | 0.686 | 1.05 (0.79–1.39) | 0.739 |
| | Forehead circumference | 1.09 (0.94–1.26) | 0.259 | 1.05 (0.84–1.32) | 0.649 | 1.02 (0.79–1.33) | 0.860 |
| | Neck circumference | 0.82 (0.71–0.95) | 0.009 | 1.07 (0.84–1.37) | 0.578 | 1.07 (0.81–1.42) | 0.615 |
| | Armpit circumference | 0.86 (0.74–1.00) | 0.046 | 1.24 (0.96–1.59) | 0.098 | 1.19 (0.89–1.60) | 0.232 |
| | Chest circumference | 0.83 (0.71–0.96) | 0.011 | 1.27 (0.99–1.64) | 0.063 | 1.26 (0.93–1.70) | 0.130 |
| | Rib circumference | 0.79 (0.68–0.91) | 0.002 | 1.26 (0.98–1.62) | 0.075 | 1.22 (0.90–1.64) | 0.197 |
| | Waist circumference | 0.82 (0.71–0.95) | 0.010 | 1.16 (0.90–1.49) | 0.246 | 1.21 (0.91–1.61) | 0.191 |
| | Iliac circumference | 0.80 (0.69–0.93) | 0.004 | 1.17 (0.90–1.51) | 0.233 | 1.13 (0.84–1.52) | 0.432 |
| | Hip circumference | 1.15 (0.99–1.34) | 0.066 | 1.19 (0.92–1.53) | 0.193 | 1.15 (0.87–1.53) | 0.320 |
| | WHtR | 0.68 (0.59–0.80) | <0.001 | 1.08 (0.84–1.39) | 0.549 | 1.10 (0.83–1.47) | 0.505 |
| | WHR | 0.66 (0.56–0.77) | <0.001 | 1.09 (0.86–1.39) | 0.474 | 1.18 (0.89–1.55) | 0.258 |
| Body composition | Total body water | 1.56 (1.33–1.84) | <0.001 | 1.40 (1.09–1.81) | 0.010 | 1.51 (1.13–2.03) | 0.006 |
| | Skeletal muscle mass | 1.61 (1.36–1.90) | <0.001 | 1.41 (1.09–1.82) | 0.009 | 1.51 (1.13–2.02) | 0.006 |
| | Body fat mass | 0.81 (0.70–0.94) | 0.006 | 1.09 (0.84–1.41) | 0.504 | 1.09 (0.82–1.45) | 0.566 |
| | Body fat percentage | 0.64 (0.54–0.75) | <0.001 | 0.99 (0.76–1.29) | 0.943 | 0.97 (0.73–1.30) | 0.864 |

P values were derived from logistic regression analyses with and without adjustment. BMI, body mass index; WHtR, waist-to-height ratio; WHR, waist-to-hip ratio.

Model 1. Adjusted for age.

Model 2. Adjusted for age, location, occupation, education level, marital status, smoking status, alcohol use, high-level activity, and SBP and DBP levels.

multiethnic cohort of participants, including African American, Japanese, Hispanic, Chinese, and white individuals in the U.S. They argued that although women who were at midlife tended to experience high WC or weight, WC and weight were not significantly associated with menopausal status, but aging was a risk factor for weight gain and increased WC. In Korean women [22], alcohol use, physical exercise participation, WC, height, weight, BMI, WHtR, and SBP were significantly different between the premenopausal and postmenopausal groups. However, these associations were not adjusted for in the analysis. Franklin et al. [15] investigated the changes in anthropometric indices and subcutaneous fat depots after menopause by magnetic resonance imaging (MRI) in an 8-year longitudinal study of white women in the US. They argued that BMI, WC, weight, and fat mass were not changed after menopause but that subcutaneous and visceral abdominal fat levels were increased after menopause. Danková et al. [17] examined the effect of menopausal status on anthropometric indices and body composition indices of healthy women in Slovakia and reported that although various anthropometric indices and body composition indices were significantly related to menopausal status, after adjustment for age, these associations disappeared. Our findings were consistent with the results of previous studies [3, 12, 14, 16, 17], indicating that obesity was not associated with the timing of menopause or menopausal status. Our results showed no associations between anthropometric indices or obesity and menopause. One of the reasons for the lack of association was assumed to be chronological aging because all significant associations in the crude analysis in this study disappeared in Model 1 after adjusting for age only as a covariate. These results are comparable to the results of previous similar studies showing that one of the risk factors for obesity in women who were at midlife was aging rather than menopause [3, 16, 23].

Many studies have pointed to differences in body composition according to menopausal status (aging, menopause, physical activity, and/or hormone changes) as the cause of changes in body composition or adiposity in midlife women. Sowers et al. [23] investigated the changes in body composition in premenopausal or early perimenopausal women and in Caucasian or African American women in a longitudinal study. They posited that both ovarian and chronological aging were associated with significant changes in WC and fat and skeletal muscle mass. Douchi et al. [12] assessed the contribution of menopause and aging to changes in fat mass and lean mass by DEXA in Japanese women. They reported that a decrease in total and trunk lean mass was associated with menopause but not with aging and that body fat mass and percentage (overall adiposity) were related to aging but were not related to menopausal status. Additionally, Lovejoy et al. [24] examined longitudinal changes in fat distribution and body composition indices in healthy women who were perimenopausal and Caucasian or African American and reported that aging increased subcutaneous abdominal fat and that menopause was related to an increase in total body fat and visceral abdominal fat. Janssen et al. [13] investigated the association of changes in body composition and hormones with the menopausal status of women at the Chicago site. Independent of age, they argued that postmenopausal women experienced significantly more visceral fat mass than pre- and perimenopausal women did, even after adjustment for all confounders, due to higher concentrations of bioavailable testosterone. Kanaley et al. [25] examined the association between abdominal fat distribution and menopausal status in Caucasian women aged 45–60 years and suggested that menopausal status and age were not associated with subcutaneous, total abdominal, or visceral fat mass levels but that physical activity levels were related to total abdominal and visceral fat mass levels. In a recent study, Marlatt et al. [11] examined longitudinal changes in body composition and cardiometabolic risk factors during the menopausal transition in white and black women and argued that white women showed greater increases in abdominal subcutaneous adipose tissue during the menopausal transition than black women did due to differences in sex steroid

hormone changes. In a narrative review, Marlatt et al. [26] stated that the difficulty in studying menopausal transition was due to variabilities in inherent hormone levels and confounders such as chronological aging. Additionally, they suggested that the menopausal transition was accompanied by changes in body composition, hormone levels, energy intake, and cardiovascular risk factors. From another point of view, Greendale et al. [27] investigated the hypothesis that the menopausal transition influences body composition independent of chronological aging. They examined the trajectory estimation of body composition by DEXA in a longitudinal study and reported that an increase in fat mass and a decrease in lean mass occurred simultaneously with the menopausal transition; therefore, no change in weight occurred at the onset of the menopausal transition. In a longitudinal study, Juppi et al. [28] assessed the relationship of menopausal transition with lean and skeletal muscle mass levels in 47–55-year-old Finnish women and argued that menopausal status appears to play an important role in the loss of skeletal muscle mass due to hormonal changes, independent of aging. Similarly, Sipilä et al. [29] investigated the differences in skeletal muscle mass and bone mineral density in premenopausal, early perimenopausal, late perimenopausal, and postmenopausal women and reported a linear decreasing trend in muscle mass across menopausal status groups; moreover, the skeletal muscle mass in postmenopausal women was significantly lower than that in premenopausal women. Our findings were similar to the results of previous studies [12, 17], indicating that body fat mass and percentage were more strongly associated with aging than with menopausal status because the statistical significance of these two indices disappeared after we adjusted for age in our models. Additionally, our results agreed with the results of a previous study [28, 29], suggesting that skeletal muscle mass was lower in postmenopausal women than in premenopausal women, independent of age.

The clinical implication of this study is that after menopause, it is important to consider a decrease in skeletal muscle mass and total body water rather than an increase in obesity or fat mass in any particular region of the body. Although many anthropometric indices and body composition indices were related to menopausal status in our study, these associations disappeared after adjustment for age. Similarly, previous studies have indicated that menopausal status induces a decrease in skeletal muscle mass and total body water intake, irrespective of age [28, 29]. However, until recently, the associations of menopausal status with obesity or adiposity have been debated. We think that the cause of the debate is aging [20, 23], physical activity [14, 25], and/or hormone changes [13, 26] in midlife women.

This study has several limitations. We cannot consider menopausal transition stage or perimenopause in this study because our data included only pre- and postmenopausal status. However, further studies are needed to analyze the changes in and characteristics of body composition and obesity indices in more detail, especially regarding menopausal status. This study was unable to reveal cause–effect relationships because of the cross-sectional design. Finally, there was limited accuracy in diagnosing menopausal status because these data were collected through self-reported interviews.

In summary, natural menopause is associated with and an important risk factor for several diseases. Compared with the findings of previous studies, in the present study, postmenopausal women may have experienced changes in total body water and skeletal muscle mass rather than changes in obesity indices and body fat mass. Specifically, we found that none of the anthropometric indices for measurements of obesity or adiposity were associated with menopause according to the adjusted models. Among the body composition indices measured in this study, body fat mass and body fat percentage were not associated with menopause, but total body water and skeletal muscle mass were significantly lower in postmenopausal women than in premenopausal South Korean women.

## Author Contributions

**Conceptualization:** Bum Ju Lee.

**Data curation:** Bum Ju Lee, Jaeuk U. Kim, Sanghun Lee.

**Formal analysis:** Bum Ju Lee.

**Investigation:** Bum Ju Lee.

**Methodology:** Bum Ju Lee.

**Project administration:** Jaeuk U. Kim, Sanghun Lee.

**Supervision:** Bum Ju Lee.

**Validation:** Bum Ju Lee, Jaeuk U. Kim, Sanghun Lee.

**Writing – original draft:** Bum Ju Lee.

**Writing – review & editing:** Bum Ju Lee.

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
