## [Decision Letter · Decision Letter 0]

12 Jul 2023

PONE-D-23-02361Association of menopausal status with body composition and anthropometric indices in Korean womenPLOS ONERespected Sir

Thank you for submitting your manuscript to PLOS ONE. After careful consideration, we feel that it has merit but does not fully meet PLOS ONE’s publication criteria as it currently stands. Therefore, we invite you to submit a revised version of the manuscript that addresses the points raised during the review process.

We look forward to receiving your revised manuscript.

Kind regards,

Nayanatara Arun Kumar

Academic Editor

PLOS ONE

Journal Requirements:

Additional Editor Comments:

Dear authors

Kindly do the corrections suggested by the reviewers and resubmit it .

thank you.

Reviewers' comments:

Reviewer's Responses to Questions

**Comments to the Author**

1. Is the manuscript technically sound, and do the data support the conclusions?

Reviewer #1: Partly

2. Has the statistical analysis been performed appropriately and rigorously? 

Reviewer #1: Yes

3. Have the authors made all data underlying the findings in their manuscript fully available?

Reviewer #1: Yes

4. Is the manuscript presented in an intelligible fashion and written in standard English?

Reviewer #1: Yes

5. Review Comments to the Author

Reviewer #1: 1. can the author please indicate the objective of the study clearly

2. what is the clinical implication/ impact of this study on clinical application

3. the results can be better presented by standardizing the decimal point

thank you

6. PLOS authors have the option to publish the peer review history of their article (what does this mean?). If published, this will include your full peer review and any attached files.

Reviewer #1: No

---

## [Author Response · Author response to Decision Letter 0]

7 Sep 2023

Response to Reviewers

To the Reviewers and Editor:

We appreciate the valuable comments that we received regarding our manuscript titled “Association of menopausal status with body composition and anthropometric indices in Korean women”. The manuscript was again revised by an English language editing service (American Journal Experts) to improve the clarity and readability of the English language in the paper. We have revised our manuscript in accordance with the Editors’ and Reviewers’ comments. Our revisions are listed below.

Additional Editor Comments:

Dear authors

Kindly do the corrections suggested by the reviewers and resubmit it.

Reviewer #1:

1. can the author please indicate the objective of the study clearly

Answer: As suggested by the reviewer, we have modified the objective of this study in Introduction section. The added contents are as follows:

“These conflicting results on the association between menopausal status and obesity may result in confusion. Therefore, the objective of the present cross-sectional study was to examine the association of menopausal status with anthropometric and body composition indices in South Korean women and to determine whether menopause status affects obesity or adiposity.”

2. what is the clinical implication/ impact of this study on clinical application.

Answer: We agreed with reviewer’s comment. Therefore, we have added the clinical implication of our study. The added contents are as follows:

“In this study, the clinical implication is that menopausal status is not associated with an increase of obesity or fat mass in any particular region of the body, but it is associated with skeletal muscle mass and total body water. Although many anthropometric and body composition indices were related to menopausal status, these associations disappeared after adjustment for age. Also, menopausal status induces the decrease of skeletal muscle mass and total body water, irrespective of age [28, 29]. But, until now, association of menopausal status with obesity or adiposity is still debatable. We think that the cause of the debate is due to aging [20, 23], physical activity [14, 25], and/or hormone changes [13, 26] in midlife women.” 

3. the results can be better presented by standardizing the decimal point

Answer: We agree with reviewer’s comment. Therefore, we have modified the decimal point for standardization throughout all tables and manuscript according to the recommendation of the following references.

[1]. TJ Cole. Setting number of decimal places for reporting risk ratios: rule of four. BMJ. 2015 Apr 27;350:h1845. doi: 10.1136/bmj.h1845.

[2]. European Association of Science Editors. EASE Guidelines for Authors and Translators of Scientific Articles to be Published in English. Acta Inform Med. 2014 Jun;22(3):210-7. doi: 10.5455/aim.2014.22.210-217. Epub 2014 Jun 15. PMID: 25132718; PMCID: PMC4130677.

[3]. TJ Cole. Too many digits: the presentation of numerical data. Arch Dis Child. 2015;100(7):608-9. doi: 10.1136/archdischild-2014-307149. Epub 2015 Apr 15.

We greatly appreciate your thoughtful comments, which helped to improve the manuscript.

---

## [Decision Letter · Decision Letter 1]

12 Dec 2023

PONE-D-23-02361R1Association of menopausal status with body composition and anthropometric indices in Korean womenPLOS ONE

Dear Dr. Lee,

Thank you for submitting your manuscript to PLOS ONE. After careful consideration, we feel that it has merit but does not fully meet PLOS ONE’s publication criteria as it currently stands. Therefore, we invite you to submit a revised version of the manuscript that addresses the points raised during the review process.

We look forward to receiving your revised manuscript.

Kind regards,

Nayanatara Arun Kumar

Academic Editor

PLOS ONE

Journal Requirements:

Additional Editor Comments:

Dear Authors

Please do the needful corrections as suggested by the reviewer . Wish you Good luck

Reviewers' comments:

Reviewer's Responses to Questions

**Comments to the Author**

1. If the authors have adequately addressed your comments raised in a previous round of review and you feel that this manuscript is now acceptable for publication, you may indicate that here to bypass the “Comments to the Author” section, enter your conflict of interest statement in the “Confidential to Editor” section, and submit your "Accept" recommendation.

Reviewer #2: (No Response)

2. Is the manuscript technically sound, and do the data support the conclusions?

Reviewer #2: Yes

3. Has the statistical analysis been performed appropriately and rigorously? 

Reviewer #2: Yes

4. Have the authors made all data underlying the findings in their manuscript fully available?

Reviewer #2: Yes

5. Is the manuscript presented in an intelligible fashion and written in standard English?

Reviewer #2: Yes

6. Review Comments to the Author

Reviewer #2: 1) The body composition and anthropometric indices measurement may be influenced by many confounders

which may influence the results.

2) Diabetes mellitus, dietary habits, familial factors, medications etc. can be the possible confounders which needs addressal.

3) What is the clinical implication of the study

4)The body composition and total body water may be influenced by the timing of the test due to diurnal and seasonal variations. When the study was performed and how was the inter/ intra observer bias in measurement negated?

5)Ideally the control should be the individual perse. When conducted as a case control study then age, parity matching and including a strict selection criteria are mandatory.

7. PLOS authors have the option to publish the peer review history of their article (what does this mean?). If published, this will include your full peer review and any attached files.

Reviewer #2: **Yes: **Yavana Suriya J

---

## [Author Response · Author response to Decision Letter 1]

8 Jan 2024

Response to Reviewers

To the Reviewers and Editor:

We appreciate the valuable comments that we received regarding our manuscript titled “Association of menopausal status with body composition and anthropometric indices in Korean women”. The manuscript was revised again by an English language editing service (American Journal Experts) to improve the clarity and readability of the language of the paper. We have also revised the manuscript in accordance with the editor’s and reviewers’ comments. Our revisions are listed below.

Reviewer Comments:

Reviewer 2

Reviewer #2: 1) The body composition and anthropometric indices measurement may be influenced by many confounders, which may influence the results.

Answer: We understand the reviewer’s comment that anthropometric indices are influenced by several covariates. Therefore, the covariates were considered in the adjusted models. Specifically, we adjusted for the covariates in models 1 and 2 in this study. For example, Model 2 was adjusted for age, location, occupation, education level, marital status, smoking status, alcohol use, high-level activity, and SBP and DBP levels. The detailed covariates in the models are described in the Statistical analysis subsection and footnotes in Table 2.

2) Diabetes mellitus, dietary habits, familial factors, medications etc. can be the possible confounders which needs addressal.

Answer: We agree with the reviewer’s comment. Unfortunately, in our study, there was no information on diseases as confounding variables. Therefore, it is very difficult to use diseases or medications as confounders.

3) What is the clinical implication of the study.

Answer: As suggested by the reviewer, we have modified the clinical implications in the last paragraph on page 13. The modified contents are as follows:

“The clinical implication of this study is that after menopause, it is important to consider a decrease in skeletal muscle mass and total body water rather than an increase in obesity or fat mass in any particular region of the body. Although many anthropometric indices and body composition indices were related to menopausal status in our study, these associations disappeared after adjustment for age. Similarly, previous studies have indicated that menopausal status induces a decrease in skeletal muscle mass and total body water intake, irrespective of age [28, 29]. However,”

4) Body composition and total body water may be influenced by the timing of the test due to diurnal and seasonal variations. When the study was performed and how was the inter/intra observer bias in measurement negated?

Answer: Body composition and total body water were measured at several hospitals according to the same protocol from 2020 to 2022. Measurements were performed by multifrequency bioelectrical impedance analysis (BIA) devices (Inbody BWA 2.0, Biospace Co. Ltd., Seoul, Korea), and the measurement device automatically calculates body composition values. Recruitment of subjects was carried out over 2 years. Therefore, it was difficult to consider diurnal and seasonal variations in automatic measurements, as in many previous studies. To provide clearer information about body composition measurements, we have modified the contents on body composition measurements in the Anthropometry and body composition measurements subsection. The modified contents are as follows:

“Body composition indices, including total body water, skeletal muscle mass, body fat mass, and body fat percentage, were measured by multifrequency bioelectrical impedance analysis (BIA) devices (Inbody BWA 2.0, Biospace Co. Ltd., Seoul, Korea) while the subjects laid on a bed with clamp-type sensors on both their wrists and ankles according to the measurement protocols and the procedures provided by the manufacturers. After resting for 5 minutes, the device began to measure and automatically calculate body composition.”

5) Ideally the control should be the individual perse. When conducted as a case control study then age, parity matching and including a strict selection criteria are mandatory.

Answer: This was a cross-sectional study rather than a case‒control study. We did not mention the prevalence of menopause because it is difficult to consider menopause a disease. Additionally, since the subjects were divided into postmenopausal and premenopausal women, it was difficult to match them according to age. We described the strict selection and exclusion criteria in the second paragraph in the Subjects and Data Source subsection and Figure 1.

We greatly appreciate your thoughtful comments, which have helped us improve the manuscript.

---

## [Editor Report · Decision Letter 2]

22 Jan 2024

Association of menopausal status with body composition and anthropometric indices in Korean women

PONE-D-23-02361R2

Dear Dr. Bum Ju Le

We’re pleased to inform you that your manuscript has been judged scientifically suitable for publication and will be formally accepted for publication once it meets all outstanding technical requirements.

Kind regards,

Nayanatara Arun Kumar

Academic Editor

PLOS ONE

Additional Editor Comments (optional):

Dear Authors

The manuscript “Association of menopausal status with body composition and anthropometric indices in Korean women” can be accepted with full consideration, All the auhtors have done their best in addressing the queries and editing the manuscript and I am really sorry for the delay in the process. I congratulate each one of you . May you do more insightful research and wishing you all the very best and god bless you all .

With respect and sincere gratitude

With warm wishes

D. Nayanatara Arun kumar

Associate Professor in Physiology

Kasturba Medical College, Mangalore
---

## [Editor Report · Acceptance letter]

8 May 2024

PONE-D-23-02361R2 

PLOS ONE

Dear Dr. Lee, 

I'm pleased to inform you that your manuscript has been deemed suitable for publication in PLOS ONE. Congratulations! Your manuscript is now being handed over to our production team.

Kind regards, 

on behalf of

Dr. Nayanatara Arun Kumar 

Academic Editor

PLOS ONE